# Circulating Levels of PD-L1, TIM-3 and MMP-7 Are Promising Biomarkers to Differentiate COVID-19 Patients That Require Invasive Mechanical Ventilation

**DOI:** 10.3390/biom12030445

**Published:** 2022-03-14

**Authors:** Leslie Chavez-Galan, Andy Ruiz, Karen Martinez-Espinosa, Hiram Aguilar-Duran, Martha Torres, Ramces Falfan-Valencia, Gloria Pérez-Rubio, Moises Selman, Ivette Buendia-Roldan

**Affiliations:** Instituto Nacional de Enfermedades Respiratorias Ismael Cosio Villegas, Mexico City 14080, Mexico; lchavez_galan@iner.gob.mx (L.C.-G.); doreydna@gmail.com (A.R.); karen.mtz92@gmail.com (K.M.-E.); hiram-ad@outlook.com (H.A.-D.); marthatorres98@yahoo.com (M.T.); dcb_rfalfanv@hotmail.com (R.F.-V.); glofos@yahoo.com.mx (G.P.-R.); mselmanl@yahoo.com.mx (M.S.)

**Keywords:** biomarkers, invasive mechanical ventilation, COVID-19

## Abstract

Background: COVID-19 is an infectious disease caused by the severe acute respiratory syndrome coronavirus-2 (SARS-CoV-2). Many COVID-19 patients require invasive mechanical ventilation (IMV) while others, even with acute respiratory failure, do not (NIMV). Therefore, we aimed to evaluate serum levels of MMP-7 and molecules related to exhausted T-cells as potential biomarkers to differentiate between IMV and NIMV patients. Methods: 105 patients diagnosed with COVID-19 and confirmed by RT-PCR for SARS-CoV-2 were divided into two groups according to the requirement for IMV. Serum levels of sPD-L1, sPD-L2, sTIM-3, sGal-9 and sMMP-7 were quantified by ELISA and correlated with clinical data. Twelve patients were followed up after eight months to compare the levels of the biomarkers between acute disease and post-COVID-19. Results: IMV patients experienced a lower PaO_2_/FiO_2_ (*p* < 0.0001) and a longer hospital stay (*p* < 0.0001), and exhibited higher levels of sPD-L1 (*p* < 0.05), sTIM-3 (*p* < 0.01) and sMMP-7 (*p* < 0.0001) when compared with NIMV patients. According to a ROC analysis, sMMP-7 had the highest sensitivity (78%) and specificity (76%) with a cut point of 4.5 ng/mL, followed by sTIM-3 and sPD-L1. Eight months post-COVID-19, IMV patients displayed a significant decrease in the initially high levels of sPD-L1, sTIM-3 and sGal-9, while sPD-L2 was increased, and sMMP-7 was unchanged. Conclusion: Circulating levels of sPD-L1, sTIM-3 and sMMP-7 are potential biomarkers of disease severity to distinguish patients requiring IMV. MMP-7 could also be a marker for the persistence of lung lesions post-COVID-19.

## 1. Introduction

COVID-19 is an infectious disease caused by the severe acute respiratory syndrome coronavirus-2 (SARS-CoV-2). The World Health Organization (WHO) declared COVID-19 as a pandemic in March 2020, and on 30 June 2021, reported 180,492,131 COVID-19 accumulated cases and 3,916,771 deaths worldwide [1].

It has been widely described that numerous COVID-19 patients develop an excessive inflammatory response, both at the local and systemic level, resulting in the patient’s deterioration [2]. Severe pneumonia and acute respiratory distress syndrome (ARDS) are common complications in these patients, and the severity of respiratory failure increases mortality by around 50%. Many of these patients require invasive mechanical ventilation (IMV), while others require only supplemental oxygen through a nasal cannula (NIMV) [3].

The development of an excessive inflammatory process in COVID-19 is likely associated with a dysregulated immune response. T cells are central players in the adaptive immune response, and within the context of COVID-19, it has been reported that critically severe patients had significantly decreased counts for CD4+ and CD8+ lymphocyte subpopulations [4]. The immune response needs modulators to maintain homeostasis, but an excess may also induce a dysregulated response. For instance, T cells from severe COVID-19 patients show an increased expression of immune-inhibitory molecules, such as programmed cell death protein ligand-1 (PD-L1) and T-cell immunoglobulin and mucin-domain containing-3 (TIM-3). It has been suggested that increased PD-L1 expression leads to CD8+ T-cell exhaustion and may contribute to the development of a cytokine storm [5,6].

In the general context of viral infection, increased levels of soluble forms of PD-L1 (sPD-L1) and TIM-3 (sTIM-3) have been reported as potential immune biomarkers related mainly to the inflammatory response [7,8]. A recent study with a small cohort of COVID-19 patients revealed that these patients displayed higher serum levels of PD-L1 than healthy controls, and it seems to have a prognostic role [9].

In other research, a high level of matrix metalloproteinase 7 (MMP-7) has been reported as an inflammatory marker in viral infection [10]. MMP-7 is a protease that, among other functions, breaks down the extracellular matrix deposited in the lung after injury, and one study has suggested that it could be a novel biomarker for COVID-19 ARDS recovery [11].

In COVID-19 patients, there is limited data on the circulating levels of sPD-L2, which like PD-L1, suppresses T cells activation. Likewise, data is lacking for TIM-3 and one of its more important ligands, the soluble form of Galectin-9 (sGal-9). Moreover, studies have yet to identify possible serum biomarkers that will help discern if a patient with COVID-19 will require invasive ventilatory support. Therefore, this study aimed to evaluate if MMP-7 and the serum level of molecules related to exhausted T-cells are increased in those patients that require IMV and can differentiate between IMV and NIMV patients.

## 2. Materials and Methods

### 2.1. Study Population

We conducted a cross-sectional study after enrolling 105 patients diagnosed with COVID-19 and confirmed by RT-PCR for SARS-CoV-2 that had been hospitalized at the Instituto Nacional de Enfermedades Respiratorias (INER), Mexico City, from March to August 2020. Twelve patients were re-evaluated eight months after they had been discharged due to improvement. Patients were divided into two groups, IMV and NIMV, according to the requirement for invasive mechanical ventilation (NIMV patients received supplemental oxygen by nasal cannula). In addition, a control group of age-matched healthy donors (HD) was selected from a previously described cohort [12]. All were asymptomatic respiratory volunteers invited to participate in our “Lung Ageing Program”, and their blood samples had been obtained prior to the COVID-19 pandemic (*n* = 23, age 51 ± 9).

### 2.2. Ethical Approval

This protocol was approved by the ethical committee of the Instituto Nacional de Enfermedades Respiratorias Ismael Cosío Villegas (Protocol numbers C41-20 and C42-20). All individuals signed a consent letter to participate in this study.

### 2.3. Clinical Data and Blood Sampling

Clinical records were reviewed to identify comorbidities, demographic data, smoking status and respiratory parameters at hospital admission. The body mass index (BMI) was used to define obesity (>30), and PaO_2_/FiO_2_ was established with arterial blood.

The blood samples for initial clinical analysis were collected at hospital admission by clinical laboratory staff using serum separator tubes containing a clot activator and a serum separator gel (BD Vacutainer 367864 or BD Vacutainer, SST). All sera were stored at −20 °C until its use.

### 2.4. ELISA Sandwich Assay

sPD-L1 (Cat. No. DB7H10, sensitivity 4.5 pg/mL), sPD-L2 (Cat. No. DY1224, sensitivity 93.8 pg/mL), sTIM-3 (Cat. No. DY2365, sensitivity 62.5 pg/mL), sGal-9 (Cat. No. DY2045, sensitivity 93.8 pg/mL) and sMMP-7 (Cat. No. DMP700, sensitivity 0.094 ng/mL) were measured with kits provided by R&D Systems (Minneapolis, MN, USA). All of the potential biomarkers were quantified by an Enzyme-Linked Immunosorbent Assay (ELISA) in sera by comparison with the corresponding standard curve and following the manufacturer’s instructions.

### 2.5. Statistical Analysis

Data were collected in Excel and analysed with Stata 13.1 (Stata Corp LP, College Station, TX, USA), GraphPad Prism 9 (GraphPad Software, La Jolla, CA, USA) and R Studio IDE 1.4.1103, RStudio, Inc. A Kolmogorov–Smirnov test was used to assess normality; parametric statistics were evaluated for those variables with a normal distribution, while for those with a free distribution, non-parametric statistics were evaluated.

Clinical and ELISA data is presented as the mean +/− standard deviation (SD). To analyse quantitative clinical variables between two groups, a Student’s *t*-test was used. In addition, the Pearson’s chi-square or Fisher’s exact test were performed as appropriate in a contingency table for clinical qualitative variables. ELISA data were analysed with a Mann–Whitney test to compare the results between two groups (HD vs. COVID-19 patients), while the comparison among three groups (HD, IMV and NIMV) was determined by an ANOVA test with a Kruskall–Wallis test to adjust for multiple comparisons.

Spearman’s correlation coefficient was used compare biomarkers with relevant demographic variables. A receiver operating characteristics (ROC) curve was generated to determine cut points, sensitivity, specificity, and the area under the curve (AUC) for the biomarkers.

## 3. Results

### 3.1. Patient Demographics and Clinical Features

The clinical characteristics and demographics of the IMV group (*n* = 76) and the NIVM group (*n* = 29) are shown in Table 1. Patients in the IMV group experienced a lower PaO_2_/FiO_2_ (*p* < 0.0001) and a significantly longer hospital stay (*p* < 0.0001). These patients also exhibited increased levels of C reactive protein (*p* = 0.0002), D-dimer (*p* = 0.0009), procalcitonin (*p* < 0.0001), lactate dehydrogenase (*p* < 0.0001), troponin (*p* = 0.002) and leukocytes (*p* = 0.001). Of note, almost all patients in the NIMV group were current smokers (*p* < 0.0001). There were no significant differences for the other variables studied.

### 3.2. sPD-L1 Is Increased in COVID-19 Patients, Mainly in Those That Required IMV

COVID-19 patients exhibited a three-fold increase in the level of sPD-L1 compared to HD (150 ± 76 vs. 52 ± 15 pg/mL, *p* < 0.0001) (Figure 1a). When the patients were divided between IMV and NIMV, we observed that IMV patients had significantly greater levels of sPD-L1 compared to NIMV patients (160 ± 84 vs. 121 ± 34 pg/mL, *p* < 0.05) (Figure 1b). Regarding the levels of sPD-L2, no differences were observed when COVID-19 patients were compared to HD (4143 ± 4059 vs. 4017 ± 1261 pg/mL) (Figure 1c), or when NIMV patients were compared to IMV patients (5128 ± 5094 vs. 3782 ± 3743 pg/mL) (Figure 1d).

### 3.3. sTIM-3 Is Increased in COVID-19 Patients, Mainly in Those That Required IMV

To ascertain if biomarkers associated with the Galectin-9/Tim-3 signalling pathway were increased in COVID-19 compared with healthy controls, and could distinguish between IMV and NIMV patients, we measured the concentration of both molecules for the three groups. We observed that COVID-19 patients had a near five-fold increase in the level of sTIM-3 compared to HD (3198 ± 1512 vs. 697 ± 255 pg/mL, *p* < 0.0001) (Figure 2a). Furthermore, and similar to our observation for sPD-L1, IMV patients had significantly higher serum concentrations of sTIM-3 than NIMV patients (3431 ± 1541 vs. 2566 ± 1249 pg/mL, *p* < 0.01) (Figure 2b).

In contrast, although the level of sGal-9 was two-fold higher in COVID-19 patients when compared to HD (8152 ± 4119 vs. 4023 ± 1414 pg/mL, *p* < 0.0001) (Figure 2c), there was no difference between the IMV and NIMV groups (8450 ± 4515 vs. 7146 ± 2499 pg/mL) (Figure 2d).

Taken together, our data indicates that two immune checkpoint pathways, PD1/PD-L1 and TIM-3/Gal-9, are altered in COVID-19, and that specifically, sPD-L1 and sTIM-3 serum levels can differentiate between IMV and NIMV patients.

### 3.4. The sMMP-7 Level Is Increased Only in COVID-19 Patients That Required IMV

It was recently reported for a small cohort of COVID-19 patients that several MMPs, including MMP-7, increase with the severity of COVID-19 (11). Here, we evaluated the level of sMMP-7 and observed a significant increase for COVID-19 patients when compared to HD (6.4 ± 4.4 vs. 4.3 ± 1.4 ng/mL, respectively; *p* < 0.05) (Figure 3a). Remarkably, we found that the level of this enzyme was increased only in those patients that required invasive mechanical ventilation, since the level for the NIMV group was similar to HD (IMV, 7.3 ± 4.5; NIMV, 4.2 ± 3.1; HD, 4.3 ± 1.4 ng/mL; IMV versus NIMV, *p* < 0.0001) (Figure 3b). Thus, our data indicate that sMMP-7 could be used as a potential biomarker to differentiate between IMV and NIMV.

### 3.5. sMMP-7, sTIM-3 and sPD-L1 Levels Can Be Potential Biomarkers to Identify IMV Patients

A receiver operating characteristic (ROC) curve analysis was applied to investigate the predictive value of the five putative biomarkers for distinguishing patients at risk of requiring IMV. According to the ROC curve, the biomarker that presented the highest sensitivity and specificity was sMMP-7 (78% and 76%, respectively; cut point 4.5 ng/mL), followed by sTIM-3 (70% and 58%, respectively; cut point 2626 pg/mL) and sPD-L1 (71% and 48%, respectively; cut point 116.83 pg/mL) (Figure 4). sPD-L2 (44% and 34% respectively; cut point 19.56 ng/mL; AUC 0.43) and sGAL-9 (53% and 54%, respectively; cut point 7874.21 ng/mL; AUC 0.56) exhibited both low sensitivity and specificity (unplotted data).

### 3.6. Correlations for Biomarkers, Serum Levels, and Clinical Characteristics

We examined whether the concentrations of the biomarkers correlated with the clinical findings for both IVM and NIMV patients. According to the data in Figure 5, we found a positive correlation between sTIM-3 and age (Rho 0.37, *p* = 0.0008), IMV days (Rho 0.31, *p* = 0.006), D-dimer (Rho 0.26, *p* = 0.02), troponin (Rho 0.39, *p* = 0.007) and procalcitonin (Rho 0.25, *p* = 0.04), and a negative correlation with PaO_2_/FiO_2_ (Rho −0.23, *p* = 0.04). MMP-7 exhibited a positive correlation with age (Rho 0.23, *p* = 0.03), IMV days (Rho 0.37, *p* = 0.001) and procalcitonin (Rho 0.33, *p* = 0.007). Interestingly, we observed a negative correlation between sPD-L2 and age (Rho −0.28, *p* = 0.01), IVM days (Rho −0.43, *p* = 0.0002), platelets (Rho −0.25, *p* = 0.05) and troponin (Rho −0.26, *p* = 0.03), and a positive correlation with PaO_2_/FiO_2_ (Rho 0.22, *p* = 0.05) (Figure 5). When we evaluated the correlations for the clinical data of interest, we detected a negative correlation between age and PaO_2_/FiO_2_ and IMV days and PaO_2_/FiO_2_, and a positive correlation between age and IMV days and IMV days and procalcitonin.

In general, these data suggest that sTIM-3 is associated with multiple disease severity variables, while sPD-L2 demonstrated the opposite behaviour.

When we correlated the clinical parameters with the biomarkers for the IMV group, we found positive correlations for the following associations: sTIM-3 with age (Rho 0.37, *p* = 0.0008), troponin (Rho 0.39, *p* = 0.007), procalcitonin (Rho 0.25, *p* = 0.04) and D-Dimer (Rho 0.26, *p* = 0.02); sMMP-7 with age (Rho 0.23, *p* = 0.0007) and procalcitonin (Rho 0.33, *p* = 0.007); sPD-L2 with PaO_2_/FiO_2_ (Rho 0.33, *p* = 0.05). There was a negative correlation for the following associations: sTIM-3 with PaO_2_/FiO_2_ (Rho −0.23, *p* = 0.04); sPD-L2 with age (Rho −0.28, *p* = 0.01), platelets (Rho −0.25, *p* = 0.05) and troponin (Rho −0.23, *p* = 0.03). On the other hand, the only correlations that we found in the NIMV group were sMMP-7 with D-Dimer (Rho 0.51, *p* = 0.007) and sPD-L2 with leucocytes (Rho −0.54, *p* = 0.005).

### 3.7. High Levels of sPD-L1 and sTIM-3 Decrease While sMMP-7 Remains Elevated after 8 Months of Recovery in IMV-Treated COVID-19 Patients

Since our data indicated that the increased levels of sPD-L1, sTIM-3 and sMMP-7 might be helpful as biomarkers for identifying IMV patients, we decided to examine whether these levels were normalised over the long-term after the patients had been discharged. Therefore, 12 IMV patients were followed up and this panel of molecules was evaluated 8 months post-COVID-19. All of these patients displayed a marked clinical, physiological, and radiological improvement, but persistent lung lesions characterised by organising pneumonia were detected with computed tomography. Notably, seven of these patients had reticular opacities and parenchymal fibrotic bands suggestive of a fibrotic response.

The data revealed that the high level of sPD-L1 had decreased around 60% 8 months post-infection (from 191 ± 109 to 80 ± 24 pg/mL; *p* < 0.0001) (Figure 6a). Likewise, we observed that the high level of sTIM-3 had decreased around 60% (from 4374 ± 2360 to 1882 ± 722 pg/mL; *p* < 0.01) (Figure 6b), and a similar profile was observed for the high level of sGal-9 (from 7082 ± 3807 to 3200 ± 1583 pg/mL; *p* < 0.01) (Figure 6c). In sharp contrast, we found that the levels of sPD-L2 had increased by more than 300% (from 1021 ± 674 to 3369 ± 2657 pg/mL, *p* < 0.01) (Figure 6d).

Interestingly, serum levels of sMMP-7 were not modified at 8 months post-COVID-19 (7.3 ± 4.8 and 8.1 ± 4.3 ng/mL, respectively) (Figure 6e). This finding indicates that, in addition to being a good candidate as a biomarker for disease severity (identifying patients that will require IMV), sMMP-7 could be a marker for persistent lung lesions post-COVID-19, since it was still elevated several months after hospital discharge.

## 4. Discussion

The SARS-CoV-2 infection causes severe disease and remains a world health concern. The disease affects mainly the lungs and may progress to pneumonia and ARDS. Given the profound severity of the symptoms and the development of respiratory failure, many patients require an extended stay in an intensive care unit, and often need to be intubated.

Our results revealed that sPD-L1, sTIM-3 and sMMP-7 levels are significantly increased in patients with severe COVID-19, and these markers could be helpful for distinguishing patients that need IMV from those who do not. Although these markers cannot be considered the only parameters for this decision, they can be combined with a set of clinical, hemodynamic and other data to help physicians decide when to intubate a patient.

PD-L1 and TIM-3 are involved in the regulation of the immune response. The PD-L1/PD-1 axis induces peripheral immune tolerance, establishing a negative feedback loop through T-effector cell exhaustion; this axis also favours the conversion of T-effector cells into regulatory T cells [13,14]. An excess of PD-L1 has adverse effects on health; for instance, the aberrant activation of the PD-1/PD-L1 pathway is a major cause of immune paralysis in patients with sepsis and other severe inflammatory responses. In this regard, reports have indicated that alleviating the inflammatory process is necessary for inducing downregulation of PD-L1 [15,16].

The expression of sTIM-3, another immune regulator, has been reported modified during COVID-19 [17]. The TIM3/Gal-9 axis mainly regulates the inflammatory process mediated by interferon (IFN)γ-producing CD4+ and CD8+ T cells [18]. In our study, the level of sTIM-3 was associated with the severity of COVID-19. It was also negatively correlated with PaO_2_/FiO_2_ and positively correlated with D-Dimer, troponin and procalcitonin. Interestingly, procalcitonin has been associated with sepsis and thrombotic diseases, suggesting that those COVID-19 patients that need IMV may have increased levels of sTIM-3 and procalcitonin as a direct consequence of an uncontrolled inflammatory process, which in turn, could be associated with the thrombotic alterations that are often observed in these patients [19]. Similarly, sGAL-9 was positively correlated with sTIM-3 and sMMP-7, confirming that it plays a vital role in regulating inflammation during viral infection [10].

sPD-L1 and sTIM-3 levels were increased in both IMV and NIMV patients; however, those requiring mechanical ventilation displayed markedly higher levels than NIMV patients. These data suggest that COVID-19 patients have dysregulated immune checkpoints that contribute to an uncontrolled inflammatory process that is strongly associated with disease severity and IMV. Moreover, it is well known that hyper-inflammation and immunoparalysis can exist concomitantly, as described for sepsis [20]. Thus, severe COVID-19 patients should have more than one altered mechanism to explain their hyper-inflammatory state. Increased levels of checkpoint molecules may induce immunoparalysis in myeloid and lymphoid cells, thereby removing a regulatory or compensatory mechanism for modulating inflammation. However, since the main aim of this study was to identify serum biomarkers that can differentiate between IMV and NIMV, we did not evaluate the functions of myeloid and lymphoid cells that could be altered by the high levels of sPD-L1 and sTIM-3. Such functions include the ability of the cells to maintain active antiviral mechanisms, phagocytosis, deliver Th2 cytokines, and their proliferative capacity, among others. Thus, our findings indicate a need to develop studies for clarifying the co-existent hyper-inflammation induced by excess sPD-L1 and sTIM-3, and if this depends upon immunoparalysis, exhaustion, immunosenescence, or anergy.

Our data demonstrated that the best biomarker for predicting IMV in COVID-19 patients is sMMP-7. The level of sMMP-7 was higher only in those patients that required invasive mechanical ventilation, while the level between HD and NIMV patients was similar, and thus, the severity of the disease is related to the level of sMMP-7. MMP-7 is a multifunctional matrix metalloprotease associated with inflammatory lung injury during ARDS. It is also overexpressed in the lung microenvironment and increased in the serum of patients with several interstitial lung diseases that may evolve into fibrosis, particularly idiopathic pulmonary fibrosis [21,22,23]. Interestingly, this enzyme remained elevated for several months in IMV survivors, and was associated with some HRCT changes suggestive of fibrosis, similar to a recent report for a cohort in which early fibrotic changes were associated with higher levels of sMMP-7 [24].

Surprisingly, sPD-L2 appeared to provide a protective effect because it was negatively correlated with days of invasive mechanical ventilation, platelets and troponin, and positively correlated with PaO_2_/FiO_2_. Moreover, this biomarker had increased 3-fold in patients that had recovered and were re-evaluated several months after hospital discharge. When we analysed the ROC curve for sPD-L2, its behaviour was not similar to that of the other biomarkers; rather, it was positive when we evaluated the NIMV group with an AUC of 0.56 (data not shown). Unfortunately, while the PD-1/PD-L1 pathway has been extensively studied, PD-L2 has received less attention. This is probably because PD-L1 is produced by a variety of circulating immune cells, whereas PD-L2 expression is restricted mainly to macrophages and dendritic cells, suggesting that PDL-2 studies should focus mainly on affected tissues [25].

Gal-9 has been proposed as an immunomodulator for excessive immunological reactions by expanding regulatory T cells [26]. In this study, the level of sGal-9 was increased in COVID-19 patients compared to HD, but its performance in the ROC analysis was low. Perhaps this biomarker is related to other processes that were not analysed in this study, such as the development of the organised pneumonia pattern detected by high-resolution computed tomography.

Finally, for those IMV patients that were followed up after 8 months, sGal-9, sTIM-3, and sPD-L1 levels had decreased substantially, suggesting the recovery of the immune response. Interestingly, whereas sPD-L1 and sTIM-1 levels had decreased after the patients had recovered, the level of sMMP-7, a molecule that is a better biomarker, was still high 8 months post-COVID-19.

## 5. Conclusions

In conclusion, our findings suggest that sTIM-3 and sPD-L1 may operate in different ways at the onset of COVID-19 to enhance a dysfunctional and exaggerated inflammatory response, which may be helpful for distinguishing patients requiring IMV. However, only sMMP-7 remained elevated, likely related to lung epithelium dysfunction, prompting new questions regarding the role of sMMP-7 in the post-COVID-19 syndrome.

## Figures and Tables

**Figure 1 biomolecules-12-00445-f001:**
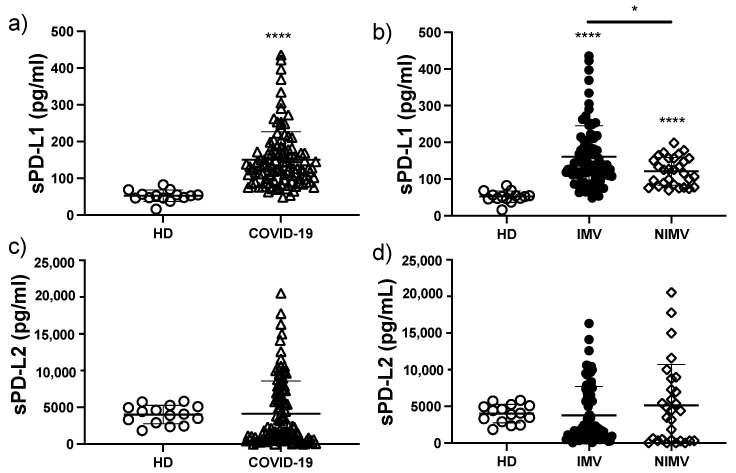
The circulating level of sPD-L1 is increased in COVID-19 patients, mainly in those that required IMV. Serum levels of sPD-L1 were evaluated in total COVID-19 patients and divided into invasive mechanic ventilation (IMV) and non-invasive mechanic ventilation (NIMV) groups (**a**,**b**). Serum levels of sPD-L2 were evaluated in total COVID-19 patients and divided into IMV and NIMV groups (**c**,**d**). Graphs show individual values and the mean ± SD. Statistical analysis was performed with the Mann–Whitney test (**a**,**c**) or an ANOVA test adjusted by the Kruskal–Wallis method (**b**,**d**). * *p* < 0.05, **** *p* < 0.0001. Asterisks indicate a significant comparison with the healthy donor (HD) group, and when the asterisk is over a line, it indicates a significant comparison between IMV and NIMV. Empty circles represent healthy donors group, empty triangles COVID-19 group, black circles IMV group, empty diamonds NIMV group.

**Figure 2 biomolecules-12-00445-f002:**
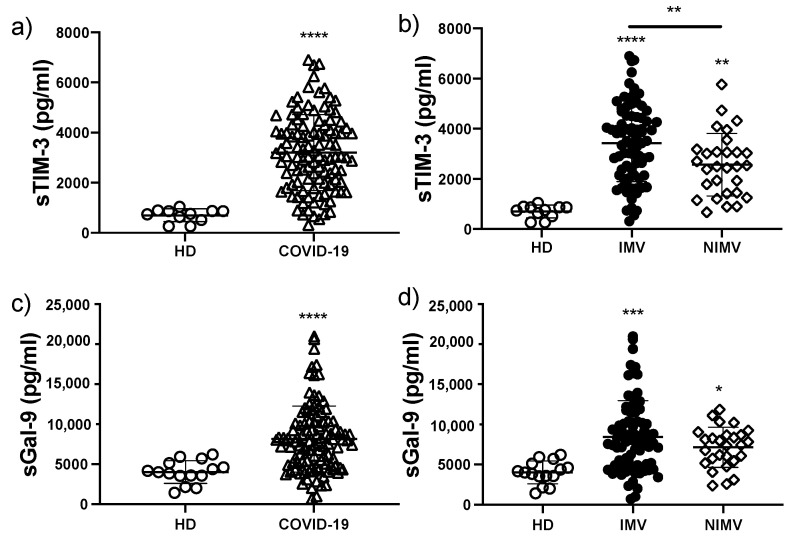
The circulating level of sTIM-3 is increased in COVID-19 patients, mainly in those that required IMV. Serum levels of sTIM-3 were evaluated in total COVID-19 patients and divided into invasive mechanic ventilation (IMV) and non-invasive mechanic ventilation (NIMV) groups (**a**,**b**). Serum levels of sGal-9 were evaluated in total COVID-19 patients and divided into IMV and NIMV groups (**c**,**d**). Graphs show individual values and the mean ± SD. Statistical analysis was performed with the Mann–Whitney test (**a**,**c**) or an ANOVA test adjusted by the Kruskal–Wallis method (**b**,**d**). * *p* < 0.05, ** *p* < 0.01, *** *p* < 0.001, **** *p* < 0.0001. Asterisks indicate a significant comparison with the healthy donor (HD) group, and when the asterisk is over a line, it indicates a significant comparison between IMV and NIMV. Empty circles represent healthy donors group, empty triangles COVID-19 group, black circles IMV group, empty diamonds NIMV group.

**Figure 3 biomolecules-12-00445-f003:**
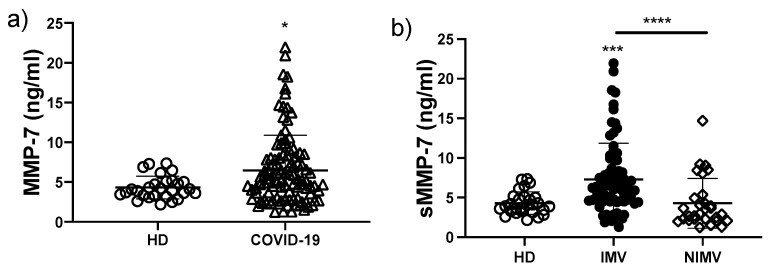
The circulating level of sMMP-7 is increased in COVID-19 patients that required IMV. Serum levels of MMP-7 (sMMP-7) were evaluated in total COVID-19 patients and divided into invasive mechanic ventilation (IMV) and non-invasive mechanic ventilation (NIMV) groups (**a**,**b**). Graphs show individual values and the mean ± SD. Statistical analysis was performed with the Mann–Whitney test (**a**) or an ANOVA test adjusted by the Kruskal–Wallis method (**b**). * *p* < 0.05, *** *p* < 0.001, **** *p* < 0.0001. Asterisks indicate a significant comparison with the healthy donor (HD) group, and when the asterisk is over a line, it indicates a significant comparison between IMV and NIMV. Empty circles represent healthy donors group, empty triangles COVID-19 group, black circles NIMV group, empty diamonds IMV group.

**Figure 4 biomolecules-12-00445-f004:**
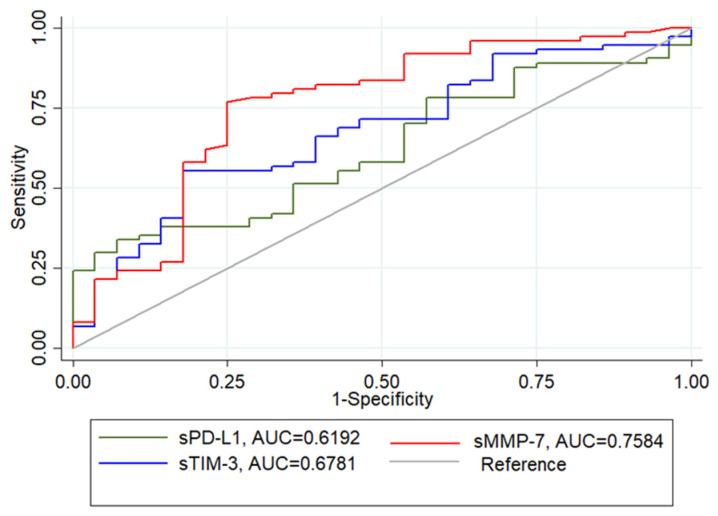
Analysis of circulating biomarkers (sMMP7, sPDL-1 and sTIM3) among IMV and NIMV using receiver operator characteristic (ROC) curves. AUC, area under the curve.

**Figure 5 biomolecules-12-00445-f005:**
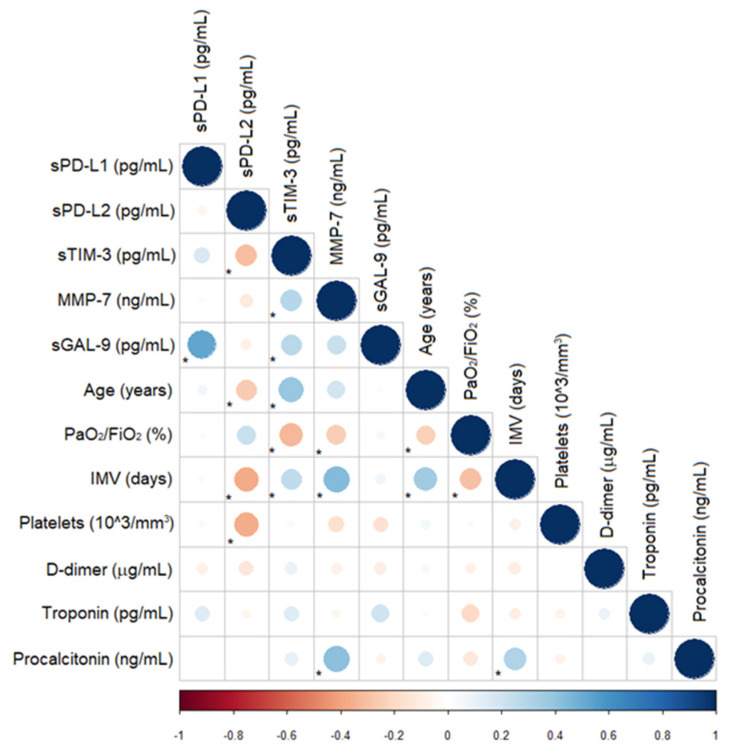
Correlogram for the association of the biomarkers with the clinical data. Blue dots represent a positive correlation, and red dots represent a negative correlation. The size and darkness of the dot are proportional to the value of the correlation coefficient. * *p* < 0.05. PaO_2_/FiO_2_, arterial pressure of oxygen/inspired fraction of oxygen.

**Figure 6 biomolecules-12-00445-f006:**
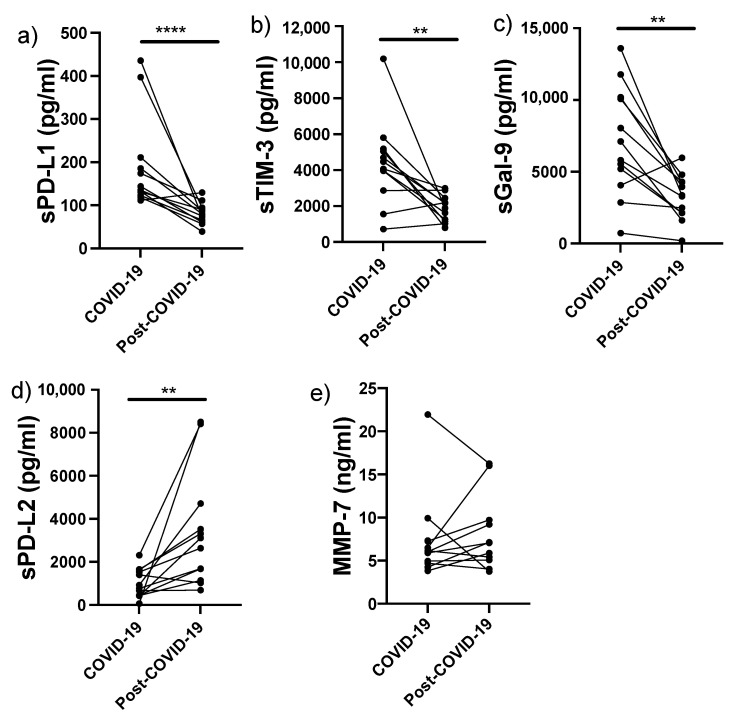
sPD-L1 and sTIM-3 levels, but not sMMP-7, were decreased 8 months after COVID-19 diagnosis. sPD-L1 (**a**), sTIM-3(**b**), sGal-9 (**c**), sPD-L2 (**d**), and sMMP-7 (**e**) were evaluated in 12 COVID-19 patients that required IMV during a follow-up 8 months after the diagnosis of COVID-19 infection (post-COVID-19). Graphs show individual values at the time of diagnosis and 8 months post-COVID-19. Statistical analysis was performed with the Mann–Whitney test. ** *p* < 0.01, **** *p* < 0.0001.

**Table 1 biomolecules-12-00445-t001:** Demographic and clinical characteristics of COVID-19 patients.

Variables	NIMV (*n* = 29)	IMV (*n* = 76)	*p*-Value
Age, years (±SD)	46 (12)	50 (11)	0.14
Male gender (%)	20 (69)	49 (64)	0.81
Body mass index (±SD)	29 (5)	29 (5)	0.72
Current smoking (%)	28 (97)	23 (30)	<0.0001
Diabetes mellitus (%)	11 (38)	21 (27)	0.36
Hypertension (%)	1 (3)	17 (22)	0.02
PaO_2_/FiO_2_ (±SD)	244 (47)	133 (64)	<0.0001
Hospital stay (±SD)	13 (11)	38 (14)	<0.0001
C-reactive protein (±SD)	8.9 (10)	24 (46)	0.0002
D-dimer (±SD)	0.87 (1)	2.39 (5)	0.0009
Procalcitonin (±SD)	0.13 (0.27)	0.62 (1)	<0.0001
Lactate dehydrogenase (±SD)	346 (131)	556 (261)	<0.0001
Troponin (±SD)	86 (163)	126 (550)	0.002
Fibrinogen (±SD)	644 (152)	732 (228)	0.07
Leukocytes (±SD)	7.6 (3)	11 (5)	0.001
Lymphocytes (±SD)	0.98 (4)	1.01 (0.75)	0.49
Platelets (±SD)	255 (154)	247 (100)	0.65

NIMV, non-invasive mechanical ventilation; IMV, invasive mechanical ventilation; PaO_2_/FiO_2_, arterial pressure of oxygen/inspired fraction of oxygen.

## Data Availability

All data relevant to the study are included in the article. The authors confirm that the raw data to support the conclusions of this study are included in the manuscript. The corresponding author will provide more information, upon rational request, to any qualified researcher.

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
