# Peer review of "Circulating Levels of PD-L1, TIM-3 and MMP-7 Are Promising Biomarkers to Differentiate COVID-19 Patients That Require Invasive Mechanical Ventilation"

_biomolecules, 2022, doi:10.3390/biom12030445_

Round 1

Reviewer 1 Report

Comments and Suggestions for Authors

The manuscript submitted by Chavez-Galan et al. presents an interesting research, where the authors evaluated soluble biomarkers to predict the severity of COVID-19 disease. Serum PD-L1, PD-L2, TIM-3, Gal-9, and MMP-7 levels were assessed to predict disease severity as defined by the need of invasive mechanical ventilation during hospitalization. They found significantly higher MMP-7 levels in patients, who required invasive mechanical ventilation compared to healthy controls and those COVID-19 patients, who received supplemental oxygen through nasal cannula. Furthermore, serum MMP-7 at 8th months after COVID-19 remained high in patients with persistent lung lesions. In addition, serum levels of PD-L1 and TIM-3 were also higher in those patients who needed invasive mechanical ventilation.

Overall, the study revealed relevant results; however, there are several comments/questions to address before publication.

  • The authors should use marker names (e.g. TIM-3 or TIM3; MMP-7 or MMP7) uniformly. The same is for COVID19 / COVID-19.

  • Authors referred normal levels of markers from their formerly published study (row 81) (Machahua et al. 2021). The referred publication includes more than one cohort as follows: “Respiratory asymptomatic volunteers aged 60 or older have been invited to participate in our ‘Lung Ageing Programme’. From this cohort, 15 individuals with ILA and 21 age-matched controls were evaluated. For comparison, we also included 28 healthy young subjects.”

This raises the question which of these groups were used in the present study as a control group. The referred study included a younger and an elderly control group with ages of 24±4 and 67±6, which are obviously different from the recent COVID-19 cohorts of 46±12 and 50±11 years for NIMV and IMV groups, respectively. As some of the markers such as MMP-7 and TIM-3 showed a significant correlation with patients’ age this aspect is critical. Therefore, please specify the number of patients with median age (+SD) for the control group. Please provide patients’ and controls’ ages as median throughout the manuscript.        

  • Description of Figure 3 is not consistent with the presented significance levels in Fig3A (one asterisk represents p<0,05...).

  • In chapter 3.5, authors stated that according to the ROC analysis 3 of 5 markers provided a better sensitivity and specify for the discrimination between NIMV and IMV. Thus, only the MMP-7, PD-L1, and TIM-3 were shown on Fig.4. Please add at least to the text the sensitivity and specificity for PD-L2 and GAL-9 also.

  • Figure 5 represents the correlation between markers and clinical data. Please add all significantly different parameters between IMV and NIMV (as listed in table 1) to this figure.

Author Response

  • The authors should use marker names (e.g. TIM-3 or TIM3; MMP-7 or MMP7) uniformly. The same is for COVID19 / COVID-19.

R= Thank you for your comment, we did it. Highlight

  • Authors referred normal levels of markers from their formerly published study (row 81) (Machahua et al. 2021). The referred publication includes more than one cohort as follows: “Respiratory asymptomatic volunteers aged 60 or older have been invited to participate in our ‘Lung Ageing Programme’. From this cohort, 15 individuals with ILA and 21 age-matched controls were evaluated. For comparison, we also included 28 healthy young subjects.”

This raises the question which of these groups were used in the present study as a control group. The referred study included a younger and an elderly control group with ages of 24±4 and 67±6, which are obviously different from the recent COVID-19 cohorts of 46±12 and 50±11 years for NIMV and IMV groups, respectively. As some of the markers such as MMP-7 and TIM-3 showed a significant correlation with patients’ age this aspect is critical. Therefore, please specify the number of patients with median age (+SD) for the control group. Please provide patients’ and controls’ ages as median throughout the manuscript.       

R= Thank you, in line 80 we described (line 80) that the HD group was age-matched, to avoid confusion and be clearer, we added the mean of age of the HD group (line 83).

  • Description of Figure 3 is not consistent with the presented significance levels in Fig3A (one asterisk represents p<0,05...).

R= We apologize for the mistake; it was corrected in the new version.

  • In chapter 3.5, authors stated that according to the ROC analysis 3 of 5 markers provided a better sensitivity and specify for the discrimination between NIMV and IMV. Thus, only the MMP-7, PD-L1, and TIM-3 were shown on Fig.4. Please add at least to the text the sensitivity and specificity for PD-L2 and GAL-9 also.

R= We add this information, row 199 and 201.

  • Figure 5 represents the correlation between markers and clinical data. Please add all significantly different parameters between IMV and NIMV (as listed in table 1) to this figure.

R= Thanks for this suggestion. To improve our manuscript, we added in the text the significantly different parameters between IMV and NIMV. Row 221-228.

Reviewer 2 Report

Chavez-Galan et al present interesting data about serum biomarkers that are increased in COVID-19 patients who require mechanical ventilation. The topic is of relevance, but I would like to point some concerns outlined below.

  1. It seems that patients with more severe disease, as those who were on mechanical ventilation had higher sPD-L1, sTIM-3 and MMP-7 levels, and authors stated that “…they could be helpful to distinguish patients who needed IMV and those who did not”. This sentence is an assumption that is not justified by the data provided since the need or not for IMV is based on physiological gas exchange values.
  2. It is interesting the role of MMP-7 in the pathogenesis of interstitial lung diseases, but again it is an assumption that this is a biomarker to predict IMV in COVID-19 patients instead it is more plausible that severity of coronavirus disease is related to MMP-7 levels.
  3. A point that deserves some comment is the role of circulating serum biomarkers of fibrogenesis (as MMP-7) and fibrotic sequelae in survivors of severe SARS-CoV-2 pneumonia, as it has been recently published (Arch Bronconeumol. 2022 Feb; 58(2): 142–149. Published online 2021 Sep doi: 10.1016/j.arbres.2021.08.014)

Author Response

  1. It seems that patients with more severe disease, as those who were on mechanical ventilation had higher sPD-L1, sTIM-3 and MMP-7 levels, and authors stated that “…they could be helpful to distinguish patients who needed IMV and those who did not”. This sentence is an assumption that is not justified by the data provided since the need or not for IMV is based on physiological gas exchange values.

R= By this comment, we re-write this sentence to explain that it could be a compliment but no exclusive parameter to decide the use of invasive mechanical ventilation (row 271-274).

  1. It is interesting the role of MMP-7 in the pathogenesis of interstitial lung diseases, but again it is an assumption that this is a biomarker to predict IMV in COVID-19 patients instead it is more plausible that severity of coronavirus disease is related to MMP-7 levels.

R= Following the suggestion, we re-write this sentence (row 315).

  1. A point that deserves some comment is the role of circulating serum biomarkers of fibrogenesis (as MMP-7) and fibrotic sequelae in survivors of severe SARS-CoV-2 pneumonia, as it has been recently published (Arch Bronconeumol. 2022 Feb; 58(2): 142–149. Published online 2021 Sep doi: 10.1016/j.arbres.2021.08.014)

R= Thank you, in the new version we included the relevant results provided by this manuscript and this reference was added (new reference #24, line 321-322).

Reviewer 3 Report

Leslie et al have looked at levels of circulating immune mediators in COVID19 patients and identified PD-L1, TIM3 and MMP-7 as useful biomarkers. Study is of importance considering the ongoing pandemic. I have some minor comments and upon addressing these, manuscript should be acceptable:

  1. authors should mention the sensitivity of the ELISA kits used.
  2. Authors should mention about 12 patients samples after 8 months time interval. Did they get COVID again? Vaccination status or any other clinical condition at the time of re-sampling?
  3. for fig.1, did patients having very high levels of sPD-L1 (>200) show any differences in clinical presentation/severity/age/disease course as compared to rest of the cohort? authors should look and reflect on this.

Author Response

  1. authors should mention the sensitivity of the ELISA kits used.

R= We add this information in methods section, row 97-99.

  1. Authors should mention about 12 patients samples after 8 months time interval. Did they get COVID again? Vaccination status or any other clinical condition at the time of re-sampling?

R= These points are very important; 12 patients did not present infection during the follow-up, the clinician’s staff discarded it.  Regarding vaccination status, the second blood sample was obtained November 2020-April 2021, our country started the vaccination in the middle of January 2021, and during the first months, it was exclusively to people older than 70 years, so our patients' groups did not receive the vaccine in the period of study.

  1. for fig.1, did patients having very high levels of sPD-L1 (>200) show any differences in clinical presentation/severity/age/disease course as compared to rest of the cohort? authors should look and reflect on this.

R= Thank you for this comment, we look for clinical correlation data with these high levels, but we do not find any correlation.